# Commercial Yeast Strains Expressing Polygalacturonase and Glucanase Unravel the Cell Walls of Chardonnay Grape Pomace

**DOI:** 10.3390/biology11050664

**Published:** 2022-04-26

**Authors:** Anscha J. J. Zietsman, John P. Moore, Jonatan U. Fangel, William G. T. Willats, Melané A. Vivier

**Affiliations:** 1South African Grape and Wine Research Institute (SAGWRI), Department of Viticulture and Oenology, Faculty of AgriSciences, Stellenbosch University, Stellenbosch 7600, South Africa; jjv2@sun.ac.za (A.J.J.Z.); moorejp@sun.ac.za (J.P.M.); 2Department of Plant and Environmental Sciences, Faculty of Science, University of Copenhagen, 1001 Copenhagen, Denmark; jouf@novozymes.com (J.U.F.); william.willats@newcastle.ac.uk (W.G.T.W.)

**Keywords:** β-1,4-endoglucanase, endo-polygalacturonase, grape pomace, grape cell wall, genetically modified commercial wine yeast

## Abstract

**Simple Summary:**

Grape skins, usually discarded during wine making, are a valuable source of cellulose (20–50%), hemicelluloses (15–20%), lignin (17–30%) and other compounds, e.g., polyphenols, which can be used as biomaterials in the manufacturing of a variety of new products, such as bioethanol or pharmaceutical products. However, to obtain these biomaterials, the complex polysaccharides of the grape cell walls must be broken down into smaller molecules to allow the extraction of compounds. The degradation process is often performed enzymatically or hydrothermally. Microorganisms that produce the required enzymes while using this waste product as a growth medium can have interesting economic advantages. Here, we created two genetically engineered wine yeast strains that produce grape cell wall degrading enzymes. These yeasts, when grown on grape pomace, induced enzymatic structural changes to the grape cell walls. A collection of antibodies binding to the different cell wall molecules were used to monitor the impact on the cell wall structure of the enzymes, confirming increased extractability of key cell wall polymers when relatively low levels of enzymes are present, illustrating the potential to develop and optimise yeast for grape waste valorisation applications.

**Abstract:**

Industrial wine yeast strains expressing hydrolytic enzymes were fermented on Chardonnay pomace and were shown to unravel the cell walls of the berry tissues according to the enzyme activities. The yeasts produced a native endo-polygalacturonase (*Saccharomyces cerevisiae* × *Saccharomyces paradoxus hybrid*, named PR7) and/or a recombinant endo-glucanase (*S. cerevisiae* strains named VIN13 END1 and PR7 END1). The impact of the enzymes during the fermentations was evaluated by directly studying the cell wall changes in the berry tissues using a Comprehensive Microarray Polymer Profiling technique. By the end of the fermentation, the endo-glucanase did not substantially modify the berry tissue cell walls, whereas the endo-polygalacturonase removed some homogalacturonan. The recombinant yeast strain producing both enzymes (PR7 END1) unravelled the cell walls more fully, enabling polymers, such as rhamnogalacturonan-I, β-1,4-D-galactan and α-1,5-L-arabinan, as well as cell wall proteins to be extracted in a pectin solvent. This enzyme synergism led to the enrichment of rhamnogalacturonan-type polymers in the subsequent NaOH fractions. This study illustrated the potential utilisation of a recombinant yeast in pomace valorisation processes and simulated consolidated bioprocessing. Furthermore, the cell wall profiling techniques were confirmed as valuable tools to evaluate and optimise enzyme producing yeasts for grape and plant cell wall degradation.

## 1. Introduction

Grape processing regularly involves the use of enzymes, either to produce wine or for the valorisation of pomace, a waste product of the wine industry. These enzymes can be commercially produced and added during the production steps or they can originate from the microorganisms (wild or inoculated yeast and bacteria) present in the fermenting must [1,2]. Microorganisms that produce processing enzymes during the normal fermentation process would be highly advantageous [3,4] and significant efforts are dedicated to isolate and investigate the natural microbial populations found in the grape and wine environment [5] in the hope of discovering novel strains with useful characteristics. Microorganisms able to hydrolyse the polysaccharides found in the cell walls of berry tissues to promote the extraction of revenue-generating molecules such as polyphenols from pomace are also sought after, since they would reduce the cost of pomace conversion [6,7] by eliminating the need for the addition of external enzymes.

Enzymes act on substrates to yield products and these reactions are influenced by the environment, leading to optimal or sub-optimal substrate conversion and/or product formation. From this perspective, grapes and fermentations are complex mixtures of substrates and environments for enzyme reactions and it is particularly difficult to identify and pinpoint the causal relationship between enzyme effects and the quality of the final wine. Moreover, to determine the extent of berry tissue cell wall hydrolysis caused by specific enzymes it is necessary to follow the changes to this matrix (substrate) directly and not only measure the products released by the enzyme actions, particularly since a wide range of hydrolysis products can be formed from this complex substrate [8].

This was the approach taken in a recent cell wall study [9] where the impact of commercial and purified enzymes on grape pomace was evaluated by using a Comprehensive Microarray Polymer Profiling technique (CoMPP) [10]. Monoclonal antibodies and carbohydrate binding modules that are specific for epitopes on the plant cell wall were used to indicate the presence and relative abundance of certain cell wall polymers. Cell walls of berry tissues were previously shown to be high in pectin [11,12,13,14], but when grape pomace was incubated with an endo-polygalacturonase (EPG) enzyme, CoMPP analysis revealed that the cell wall composition did not change [9]. Strikingly, when the EPG enzyme was present in combination with an endo-glucanase (EG) that did impact the cell wall structure, a synergistic (larger than EG alone) cell wall degradation effect was seen. These results were insightful and provided a view of the stepwise unravelling of the berry cell walls that occurs during fermentation in the presence of enzymes.

Here we investigated how EPG and EG produced by yeast strains during a fermentation on Chardonnay pomace degrade and/or change the composition of grape berry cell walls under optimal pH conditions. After the fermentation, the cell walls of the pomace were studied with CoMPP and this was supported by monosaccharide compositional analysis, revealing variation in the composition of the cell wall due to the different treatments. Multivariate data analysis was used to detect and confirm the major patterns in the data.

## 2. Materials and Methods

### 2.1. Microbial Strains and Culture Conditions

The sources and relevant genotypes of bacterial and yeast strains, as well as plasmids used in this study, are listed in the Appendix A. *Escherichia coli* transformants were grown in Luria-Bertani (LB) broth (Biolab, North Shore, New Zealand) containing 12 g/L tryptone, 12 g/L NaCl and 6 g/L yeast extract supplemented with 100 μg/mL ampicillin (Roche, Basel, Switzerland) for plasmid selection. Yeast peptone dextrose (YPD) broth (Biolab) containing 1% yeast extract, 2% peptone and 2% glucose was used for the general culturing of yeast cells. *S. cerevisiae* transformants were cultured on yeast extract (YE) plates containing 0.5% yeast extract, 2% glucose and 3% Pastagar B (Bio-Rad Laboratories, Hercules, CA, USA) supplemented with 150 μg/mL phleomycin (Sigma-Aldrich, St. Louis, MO, USA) or 200 μg/mL G418 (geneticin) (Sigma-Aldrich).

### 2.2. DNA Manipulations and Plasmid Construction

Standard methods were used for plasmid DNA isolation, restriction enzyme digestion, ligation reactions and transformation of *E. coli* DH5α [15], unless otherwise stated. Lambda DNA and all restriction enzymes were obtained from Roche. All gel isolations of DNA fragments were performed with the Zymoclean DNA recovery kit (Zymo Research Corporation, Irvine, CA, USA) according to the manufacturer’s instructions. TaKaRa ExTaq™ DNA polymerase (Takara Bio Inc., Kusatsu, Japan) was used in polymerase chain reactions (PCR) required for the manipulation of DNA while GoTaq^®^ DNA Polymerase (Promega Corporation, Madison, WI, USA) was used for screening and confirmation PCR reactions.

The primers and the PCR programs used in this study are listed in Appendix A. PCR amplifications were performed with a thermal cycler (Hybaid PCR express, Sigma-Aldrich) and the PCR product fragments were isolated from an agarose gel, ligated into pGEM^®^-T Easy (Promega Corporation) and transformed into *E. coli* DH5α. The integrity of the PCR products and the different clones was verified by restriction analysis and sequencing (Central Analytical Facility, Stellenbosch University).

### 2.3. Construction of Integration Cassette POF1-TEF2_P_-kan^r^-TEF2_T_-POF1 (POF1-KMX)

Primers POF1-L and POF1-R (Appendix A) were designed to amplify the region from 112 bp on the 5′ side of the start codon of the *POF1* gene, up to 466 bp on the 3′ side of the stop codon. Program 1 (Appendix A) was used to obtain the 1302 bp fragment (POF1), which was cloned into pGEM^®^-T Easy yielding pPOF1. The *Sal*I site in the multiple cloning site of this plasmid was destroyed by digestion with this restriction enzyme and then filling up the 5′ protruding ends with the DNA Polymerase I (Klenow) Large fragment (Promega Corporation) and then ligating the blunt ends. The resulting plasmid was named pPOF1a and was further modified by introducing a new *Sal*I site within the coding sequence of the *POF1* gene on the plasmid. This was achieved by using primers POF1-L and POF1-R and plasmid pMPOF1b (described in the next section) as template and PCR program 1. The 1270 bp PCR fragment was cloned into the pDRIVE forming plasmid pDRIVE POF1b, whereafter the POF1b fragment (with the *Sal*I site) from pDRIVE POF1b was isolated by digestion with *Stu*I; this fragment replaced the POF1 fragment (without the *Sal*I site) in pPOF1a to yield pPOF1b. Finally, the *TEF2_P_ kan^r^ TEF2_T_* cassette was isolated from pUG6 by digestion with *Xho*I and *Bgl*II and this was cloned into the *Bam*HI and *Sal*I sites of pPOF1b to yield pPOF1b KMX (Figure 1A).

### 2.4. Construction of Integration Cassette _M_POF1-ADC1_P_-MFα1s-END1-TRP5_T_-_M_POF1 (MPOF END1)

Primers MaxiPOF1-L and -R (Appendix A) were used to amplify the region from 815 bp on the 5′ side of the start codon of the *POF1* coding sequence up to 784 bp 3′ from the stop codon. Genomic DNA from VIN13 was used as template with PCR program 2 (Appendix A). The PCR product (MPOF1) was cloned into pGEM^®^-T Easy resulting in pMPOF1. The *Sal*I site of this plasmid was destroyed in the same way as with pPOF1. This plasmid was called pMPOF1a.

The next step was to introduce a new *Sal*I site within the open reading frame of the *POF1* gene that can serve as a position to introduce the END1 expression cassette. An 846 bp PCR fragment was produced using pPOF1 as template, primer pair POF1-M and POF1-R and PCR program 3. This PCR product was cut with restriction enzymes *BamH*I and *Sna*BI and ligated into the corresponding restriction enzyme sites of pMPOF1a, yielding pMPOF1b. The final step was to isolate the 2820 bp *ADC1_P_-MFα1s-END1-TRP5_T_* fragment from pAR5 (*Sal*I) and clone this into the newly created *Sal*I site of pMPOF1b to yield pMPOF1bEND1 (Figure 1B).

### 2.5. Yeast Transformation

Plasmids pPOF1b KMX and pMPOF1bEND1 were digested with *Stu*I and *Sma*I respectively to liberate the integration cassettes from the rest of the plasmids. VIN13 and PR7 were transformed, first with the POF KMX integration cassette (from pPOF1b KMX), according to the electrophoresis protocol described in [16]. However, a few changes were made: the yeast culture was grown until the late exponential phase (OD_600_ = 0.8), the DNA integration cassette concentration was increased to 4 μg and the regeneration after the electroporation pulse was performed overnight. Putative (V13 KMX and PR7 KMX) transformants were selected from YE-G418 plates. After confirming the integration event with PCR (see next section), two transformed yeasts were selected for the next round of transformations where the POF-KMX cassette was replaced by the MPOF END1 expression cassette. This was a co-transformation that included 100 ng of the pUT332 plasmid which was used as a screening tool to limit background colonies. Putative transformants were selected from YE-phleomycin plates and transferred with sterile toothpicks to YE-G418 plates (negative screen) to select colonies whose POF KMX cassette was putatively replaced by the MPOF END1 cassette.

### 2.6. Verifying the Yeast Transformants and the Position of Integration

The integration of the POF1 KMX cassette into the POF1 locus of the VIN13 and PR7 genome was confirmed by using primers POF1-L and KanMX-Rp (PCR program 2, Appendix A) on genomic DNA isolated (according to the protocol by [17]) from colonies growing on YE-G418 plates. After replacing the POF KMX with MPOF END1 expression cassette, primers END1SEQ and MaxiPOF1-L were used with PCR program 4, to screen the putative VIN13 END1 and PR7 END1 strains. The presence of a PCR product of 1971 bp indicated that the *POF1* locus was successfully interrupted by the *END1* expression cassette (Appendix A).

The position of the integration event was further investigated by designing a primer, exPOF1-L that aligns on the yeast genome, 167 bp 5′ from the target for homologous recombination. This primer was used together with primer ENDSEQ (PCR program 5, Appendix A) to yield a band size of 2138 bp that indicated a successful integration. By using primers POF1-L and POF1-R and genomic DNA of V13 END1 transformant #1.8 and PR7 END1 transformant #4.7, we could establish whether the integration event happened only at one or both POF1 loci of these two diploid yeast strains.

### 2.7. Enzyme Activity

To confirm that the transformed yeast could produce active recombinant β-1,4-glucanase from the integrated expression cassettes, cells from VIN13 END1 and PR7 END1 colonies were transferred to carboxymethyl cellulose (CMC) plates (0.5% yeast extract, 2% glucose, 2% agar, 1% CMC) and grown for 3 days at 30 °C, whereafter the colonies were washed off with sterile distilled water, then flooded with 0.1% Congo red for two hours, followed by a de-staining step with 1 M NaCl_2_. The activity of the native polygalacturonase enzyme of PR7 and PR7 END1 strains was confirmed with a plate assay, as previously described in [18].

### 2.8. Curing of the pUT332 Plasmid and Stability of Yeast Transformants

The yeast strains were cultivated in YPD media at 30 °C for 100 generations. At the end of 100 generations the GMY (genetically modified yeast) cultures were plated on YPD medium. One hundred randomly selected colonies were analysed for the presence of the *end1* gene. Two different methods were used. Initially, yeast colonies were dissolved in 5 μL of sterile distilled water and the suspensions were transferred to QIAcard FTA cards (QIAGEN, Hilden, Germany). The cards were prepared as specified by the manufacturer and used as template in a PCR reaction to detect the presence of the *end1* gene using primers END1SEQ and exPOF1-L as previously described. Additionally, the colonies were transferred to CMC plates and tested for β-1,4-glucanase activity as previously described. Finally, the colonies were tested for sensitivity towards phleomycin by transferring them using sterile toothpicks to YE-phleomycin plates. If colonies were unable to grow on these plates, it was an indication that the yeast cells eliminated the pUT332 plasmid during the non-selective prolonged growth period.

### 2.9. Fermentation on Pomace

*Vitis vinifera* cv Chardonnay grapes were harvested in the Stellenbosch region of South Africa and the pomace was obtained after the grapes were pressed. The pomace was kept at −20 °C until processed, which entailed seed removal and blending with a food processor. The chopped/blended pomace was then prepared as a 15% (*w*/*v*, wet weight) suspension in a 0.12 M citrate phosphate buffer at pH 6 and autoclaved (121 °C, 15 min, 100 kPa) to sterilise as well as destroy the native grape and microbial enzymes. The yeast strains (VIN13, V13 END1, PR7 and PR7 END1) were pre-cultured in YPD and inoculated at a cell density of ca. 2.2 × 10^7^ cells/mL into the pomace suspension (determined with plate counts just after inoculation). The pomace cultures were fitted with fermentation caps and incubated with shaking at 30 °C for 144 h. All fermentations were conducted in quadruplicate and, after incubation, the supernatants and solids were separated by centrifugation and frozen at −20 °C until further analyses.

### 2.10. Analysis of the Supernatants of the Pomace Fermentation

Two samples per biological repeat were taken one hour after inoculation, and then at 48 h, 120 h and at 144 h (end of fermentation) in order to determine the reducing sugars as an indication of fermentation progress, using the method described by Ghose [19]. The pH of all the supernatants (the average of two technical repeats) was determined on the final day.

### 2.11. Isolation of Cell Walls or Alcohol Insoluble Residue (AIR) from Pomace Solids

The frozen pomace pellets (representing all the solid material from one biological repeat), collected at the end of the fermentation, were freeze-dried, and then dry milled and homogenised with a Retsch MM400 mixer mill (Retsch GmbH, Haan, Germany) at a frequency of 30 Hz for 30 s. The resulting powder was used for the isolation of the cell walls using the method described by Ortega-Regules et al. [20] with some changes. The first acetone step after the buffered phenol extraction was omitted. Furthermore, after the third acetone wash, the excess acetone was removed under a slight vacuum and the remaining cell walls or alcohol insoluble residue (AIR) was then mixed with an equal volume of sterile distilled water, frozen at −80 °C and then freeze-dried.

### 2.12. Monosaccharide Analysis of Cell Wall Samples

The monosaccharide composition of the AIR isolated from the pomace was analysed according to the method described in [21,22], with some modifications. Approximately 5 mg of AIR was hydrolysed (two technical repeats per biological repeat) to monosaccharides (2 M TFA, 110 °C, 2 h) followed by derivatisation to methoxy sugars at 80 °C for 16 h. After silylation with HMDS + TMCS + Pyridine 3:1:9 (Sylon HTP) kit (Supelco, Inc., Bellefonte, PA, USA), the derivatives were separated and analysed in a gas chromatograph, Agilent 6890 N (Agilent Technologies, Inc., Palo Alto, CA, USA) coupled to an Agilent 5975 MS mass spectrometer detector, using a polar (95% dimethylpolysiloxane) ZB-SemiVolatiles Guardian (30 m, 0.25 mm ID, 0.25 µm film thickness) GC column. The oven temperature was maintained at 70 °C for 2 min, ramped at 1 °C/min to 76 °C, then at 8 °C/min to 300 °C and then held for 5 min. We expressed the monosaccharide composition as the mole percentage contribution of each monosaccharide in relation to the nine common plant cell wall monosaccharides present. Error bars in the histogram represent the mean of four biological samples.

### 2.13. Comprehensive Microarray Polymer Profiling (CoMPP) Analysis of Cell Wall Fractions

One AIR sample (10 mg) from each biological repeat was used for CoMPP analysis. Sequential extractions, according to the procedure described in Moore et al. [11], were performed on these AIR samples, using the solvents CDTA (diamino-cyclo-hexane-tetra-acetic acid) and NaOH respectively. CDTA extracts mainly pectin-rich polymers from the cell wall particles and the second extraction with NaOH solubilises the hemicellulose-rich polymers. A selection of monoclonal antibodies (mAbs) and carbohydrate binding modules (CBMs) were chosen to cover the variety of epitopes expected in the grape cell wall. A full list is provided in the Appendix A. Arrays printed with the different fractions were probed individually with the mAbs and CBMs as described in Moller et al. [10]. Each sample was represented by five microarray spots and a mean spot signal was calculated, whereafter it was normalised to the highest signal (set as 100) in the dataset. A cut-off value of five was imposed on the data.

### 2.14. Multivariate and Univariate Statistics

Univariate statistical analysis (unpaired *t*-test, 95% confidence level) was performed on monosaccharide data. Multivariate analysis by means of principle component analysis (PCA) [23] was performed using SIMCA 13 software package (Sartorius Stedim Data Analytics AB, Umeå, Sweden).

## 3. Results

In this study we investigated the effect of an EPG and an EG enzyme produced by yeast strains on the cell wall composition of grape pomace that was used as fermentation substrate. To this end, we used the PR7 strain (trade name Exotics SPH, Anchor yeast, South Africa), a hybrid between *Saccharomyces paradoxus* R088 [24] and VIN13 (Anchor yeast, South Africa), a *S. cerevisiae* commercial wine strain. *S. paradoxus* R088 produces an endo-polygalacturonase (*PGU1*) [25] and PR7 inherited this gene during the hybridisation process and displays similar endo-polygalacturonase (EPG) activity to the *S. paradoxus* R088 on a plate assay. The *S. paradoxus* RO88 Pgu1 enzyme has an optimal pH of 5.5 with 43% residual activity at pH3.5 [26]. A recombinant PR7 strain (PR7 END1) that expressed a *Butyrivibrio fibrisolvens* H17c β-1,4-D-glucanase (*end1* gene) [27] from a genome-integrated cassette was engineered to yield a strain with both EPG and EG activities. Endo-β-1,4-glucanases randomly hydrolyse internal β-1,4-D-glycosidic bonds in cellulose and xyloglucan, producing oligosaccharides and reducing the polymer length. According to [28], this enzyme is irreversibly inactivated when the pH decreases below 4.5, and thus this enzyme would not be active in a grape must matrix where the pH is usually below 4 [29]. The VIN13 yeast strain was used as a negative control (no EPG or EG activity) and a VIN13 engineered to express the *end1* gene (VIN13 END1) was used to demonstrate the effect of the endo-glucanase activity on its own. The yeast strains were fermented on autoclaved Chardonnay pomace (deseeded pulp and skins) suspended in a buffer at pH 6 to ensure that both the EPG and EG expressed by the yeast strains would be active and close to their optimum pH. After the fermentation, the cell walls of the Chardonnay pomace were studied to determine the effect of the different conditions/treatments in the study.

### 3.1. Construction of Genetically Modified Yeast

Two integration cassettes were developed for this study. The first of these contained the expression cassette that conveys resistance towards geneticin/G418 (*TEF_P_ kan^r^ TEF_T_*) with flanking regions that are homologous to the *POF1* (phenylacrylic acid decarboxylase) locus on the genome of *S. cerevisiae* (Figure 1A). The second integration cassette (Figure 1B) contained extended *POF1* homologous regions and the expression cassette for the EG gene (*end*1) from *B. fibrisolvens* under control of the *S. cerevisiae ADC*1 promoter and *MFα*1 secretion signal. The larger homologous region in the MPOF END1 integration cassette was designed to ensure that the MPOF END1 integration cassette would be able to replace the first integration cassette (POF KMX) on the yeast genome in two consecutive integration events.

VIN13 transformed with the *kan^r^* integration cassette delivered more than 200 putative transformants showing resistance to G418. Genomic DNA of ten of these putative transformants (randomly chosen) was tested for the presence of the *kan^r^* integration cassette with primers POF1-L and KanMX-Rp, and nine of the transformants gave the correct PCR product of 2130 bp (Appendix A). The first transformant was selected for the subsequent co-transformation with MPOF END1 and pUT332 resulting in 1700 putative transformants. These putative transformants were screened for restored sensitivity towards G418, which would indicate that the *kan^r^* integration cassette was replaced by the MPOF END1 integration cassette. Eighteen putative V13 END1 transformants (G418 sensitive) were isolated from the master plates and transferred to CMC plates to test for *end1* enzyme activity. Nine of the eighteen putative V13 END1 transformants showed EG activity on the CMC plates (data not shown) and delivered a PCR product of 1971 bp (primer pair END1SEQ and MPOF-1 on genomic DNA isolated from the transformants) (Appendix A), indicating the presence of the END1 expression cassette at the intended position on the yeast genome.

The same procedure for the integration of the END1 integration cassette was followed for the PR7 yeast strain. Eleven PR7 KMX transformants were obtained and verified with primers exPOF1-L and KanMX-Rp (PCR product 2297 bp) (data not shown). The first positive transformant (named PR7 KMX #1) was used in the follow up transformation, which rendered 1800 putative transformants. After screening 520 of these transformants for G418 sensitivity, eleven did not grow on YE-G418 plates and eight of the eleven gave a PCR product (1971 bp) using the exPOF1-L and ENDSEQ primer pair (Appendix A). Two transformants (PR7 END1 #10.10 and #4.17) were tested on CMC plates for endo-glucanase activity and both were positive (Figure 2). The integration strategy was therefore successful in both the VIN13 and PR7 yeast strains. A PCR reaction on the genomic DNA of V13 END1 #1.8 and PR7 END1 #4.17 with primers POF1-L and POF1-R yielded PCR products of 3944 bp (data not shown) indicating that both copies of the *POF1* gene of these two diploid yeast strains [25] were disrupted. The undisrupted POF1 gene would have given a PCR product of 1274 bp.

### 3.2. Activity of the Recombinant and Native Yeast Enzymes and Stability of the Integration

The activity of the recombinant EG enzyme in the V13 END1 and PR7 END1 strains as well as the native polygalacturonase enzyme was tested by growing the strains on agar plates containing the enzyme substrates. All the PR7 strains showed clear zones inside white rings (Figure 2a), indicating an active native polygalacturonase, while VIN13 strains produced no clearing zones. Positive (light yellow to clear zones on CMC plates) zones could be seen (Figure 2b) for all the VIN13 END1 and PR7 END1 GMY (indicating EG activity), while the untransformed VIN13 and PR7 strains showed no zone.

To test the stability of the integration event the GMY were cultivated for 100 generations with no selective pressure, whereafter 100 colonies of each strain were tested by PCR and on activity plates for the presence of the recombinant gene. All strains showed 100% stability of the integration (data not shown) and their inability to grow on agar plates containing phleomycin, proving that they were cured of the pUT332 plasmid.

### 3.3. Fermentation on Chardonnay Pomace

Transformed yeast strains VIN13 END1 #1.8 and PR7 END1 #4.17 (henceforth called VIN13 END1 and PR7 END1) were chosen to use in all subsequent fermentations on pomace, together with untransformed VIN13 and PR7. The reducing sugar in the supernatant of the pomace suspension increased from 22 mg/mL to 25.9 mg/mL during the autoclaving process. The fermentation progress was followed by recording the decrease in reducing sugars on the first day (an hour after inoculation), after 48 h, 120 h and 144 h (last day) (Appendix A). All the yeast fermentations reduced the sugar within 48 h to less than 3 mg/mL and at the end of the fermentation period (144 h), the reducing sugar concentrations were 1.93 (VIN13), 1.87 (V13 END1), 2.04 (PR7) and 2.04 mg/mL (PR7 END1).

The pH of the pomace suspension after autoclaving was 5.7 and this changed to 5.9 in the control reaction, while the pH of the yeast fermentations was stable at 5.7. Thus, the recombinant EG protein that is irreversibly inactivated at pH values below 4.5 [28] would not have been negatively affected by pH during the fermentation. The same applies for the native EPG of PR7 (optimum pH of 5.5 [26]), if we assume that it has the same properties as the EPG expressed by the parent strain, *S. paradoxus* RO88.

### 3.4. Analysis of the Pomace Fermentation Residue

To determine if the hydrolytic enzymes (EG and EPG) produced by the yeast strains (V13 END1, PR7 and PR7 END1) changed the cell wall composition of the pomace, we analysed the AIR isolated from pomace residue at the end of the fermentations. The results from V13 END1, PR7 and PR7 END1 were compared with the control fermentation (no yeast present) and the VIN13 fermentation (no EG or EPG activity).

The CoMPP data (Figure 3) from the CDTA, pectin-rich fraction showed that only low levels of homogalacturonan (HG) (mAbs LM18 and LM19) were still present in the pomace cell walls after the autoclaving step (control sample), but relative high levels of unbranched rhamnogalacturonan-I (RG-I) (mAbs INRA-RU1 and INRA-RU2) could be extracted, as well as some RG-I side chains (mAbs LM5 and LM6), xyloglucan (mAb LM25), extensins (mAbs JIM11 and JIM20) and AGPs (mAbs JIM8, JIM13 and LM2). We observed the same trend due to autoclaving in a previous study [9].

We observed a significant (95% confidence interval) difference in the monosaccharide levels (Figure 4) between the unfermented (control) samples and all the fermented samples (VIN13, PR7, VIN13 END1 and PR7 END1).

The fermentation process decreased the mol% of all the monosaccharides except Man (mannose) and Glc (glucose). The latter had significantly higher levels in the fermented than in the control samples. The high mol% Glc in all the fermented samples corresponded with the strong signal for β-1,3-D-glucan (mAb BS-400-2) (Figure 3, CDTA and NaOH fraction). The BS-400-2 mAb recognise β-1,3-D-glucan, also known as callose (a polymer formed in plant tissue as a response to wounding or pathogen attack), but it could possibly also recognise the β-1,3-D-glucan that is part of the yeast cell wall. The fermented samples showed a higher abundance (Figure 3, NaOH fraction) in HG (mAbs PAM1 and LM19), RG-I (mAbs INRA-RU1 and INRA-RU2) and β-1,4-D-galactan and α-1,5-L-arabinan (mAbs LM5 and LM6).

The samples fermented with VIN13 and VIN13 END1 did not show (Figure 3) any significant cell wall changes compared to the control sample apart from the general fermentation response already mentioned, whereas the PR7 samples that produce only the native EPG displayed a decrease in HG (mAbs LM18 and LM19) labelling (Figure 3, CDTA fraction). The recombinant PR7 END1 yeast strain that produces both a native EPG and a recombinant EG seems to have an influence (Figure 3, CDTA fraction) on the pomace cell wall composition in the following ways: (i) there was a decrease in HG (mAbs LM18 and LM19), similar to that which was seen for the untransformed PR7; (ii) the PR7 END1 strain caused an increase in the labelling for unbranched RG-I polymers (mAbs INRA-RU1 and INRA-RU2); (iii) there was also a slight increase in the RG-I side chains (mAbs LM5 and LM6); and (iv) we also saw an increase in the cell wall proteins for the samples fermented with PR7 END1 (mAbs JIM11, JIM20 and JIM13).

For an overview of the CoMPP data, we used principal component analysis (PCA) plots (SIMCA 13) (Figure 5 and Figure 6), which shows the relationships between the different samples according to the abundance of the polymers in each sample.

The CDTA fraction showed (Figure 5) that the PR7 END1 samples cluster together and separate from the control samples along the first component axis (PC1 47%). The separation is driven by an abundance of RG-I side chains, unbranched RG-I and cell wall proteins in the PR7 END1 samples. PR7 and PR7 END1 separated from the VIN13 and control samples along the second component axis (PC2 13%) because of the abundance in mAbs LM18 and LM19 in the latter samples. In Figure 5, the variable mAb BS-400-2 was excluded from the analysis to see the differences between the samples without the contribution of (presumably) the yeast cell wall. The exclusion of this variable had very little effect (PC1 changed from 46% to 47% and PC2 were unchanged), and, furthermore, the samples and variables had nearly identical distribution patterns than the original score and loading plot.

In the NaOH fraction (Figure 3), the combination of EPG and EG activity in PR7 END1 showed the largest increase (compared to the other fermented samples) in RG-I and RG-I side chains; however, in contrast to the CDTA fraction, they caused no increase in the cell wall proteins in the NaOH fraction. Thus, although it was possible to see a difference between PR7 END1 and all the other fermented samples, the effect was less clear than in the CDTA fraction. A PCA plot of the CoMPP data from the NaOH fraction (that excludes mAb BS-400-2) (Figure 6) confirmed the differences seen in Figure 5 and showed the PR7 strains located on the opposite side of the plot from the control samples. The separation was driven by the abundance in RG-I and RG-I side chains. The fermented samples were separated from the controls because of an abundance in HG (mAb PAM1). Again, the exclusion of mAb BS-400-2 did not influence the PCA analysis; the general distribution pattern remained the same, while PC1 only changed from 34 to 35%, and PC2 from 20 to 18%.

## 4. Discussion

With this study we demonstrated the successful and stable integration of two copies of an expression cassette into the genome of two industrial wine yeast strains, VIN13 and PR7 (Exotics SPH). The MPOF END1 expression cassette enabled these yeast strains (V13 END1 and PR7 END1) to produce an active EG originating from a rumen bacteria [27]. After the integration and removal of the pUT332 plasmid, the only foreign DNA still present in the yeast was the *end1* gene. The POF1 genome locus (also known as PAD1) was targeted for integration at position 663 of the open reading frame that has 728 bp.

The reducing sugar concentration of about 25 mg/mL that was available in the pomace supernatant was a sufficient carbon source for the yeast strains and was quickly consumed and reduced to 2 mg/mL. No difference could be observed between the untransformed and the recombinant strains in terms of sugar consumption (Appendix A). This is also an indication that the pretreatment process, which can lead to by-product formation and inhibition of fermentative microorganisms [30], was not too severe. All the fermented samples had higher mol% of Man and Glc than the unfermented control samples. This is probably due to the high levels of Man and Glc present in the cell walls of the yeast inoculated to perform the fermentations. At the end of the fermentation, the pomace solids were separated from the supernatant by centrifugation and the yeast cells were thus collected and freeze-dried together with the pomace residue. The *S. cerevisiae* cell wall composition varies depending on the growing conditions but on average it contains 35–40% mannoproteins, 5–10% β-1,6-glucan, 50–55% β-1,3-glucan and 1–2% chitin (wall dry weight [%]) [31]. The slight differences between Glc levels of the samples fermented with VIN13 and PR7 (16.4% and 14.5% respectively) are probably due to differences in their glucan cell wall content (Moore et al., unpublished). In contrast to Man and Glc, all the other monosaccharides had reduced concentrations in the fermented pomace cell walls compared to the control samples. From previous results, we know that the levels of Ara (arabinose) in cell walls of *V. vinifera* cv Pinotage skins decrease during alcoholic fermentation [14], which is probably due to the mobilisation of arabinan and arabinogalacturonan proteins (AGPs) to the fermenting must. However, none of the other decreases were previously seen. Moreover, since the monosaccharide levels are given as a mol% relative to each other, the decrease might only be in response to the increase in Man and Glc (due to the presence of yeast cell walls).

The VIN13 strain was chosen as a fermentation control that does not secrete any of the enzymes evaluated in this study. *S. cerevisiae* produce several extracellular, intracellular and cell wall bound glucanases (*EXG1*, *BGL1*, *EXG2*, *SSG1/SPR1* and *BGL2*) but they all are β-1,3-specific glucanases and thus inactive towards the β-1,4-linked cellulose and xyloglucan of the grape berry cell wall [32]. The VIN13 and VIN13 END1 strains did not alter the cell wall composition (CoMPP results) apart from the fermentation modifications seen for all the fermented samples. In our analysis, the EG enzyme on its own was not capable of degrading noticeable amounts of cell wall polymers. Furthermore, the VIN13 strains did not decrease the mAb BS-400-2 labelling, which indicates that none of the enzymes secreted by VIN13 could degrade this specific polymer or that this polymer was not within reach of the VIN13 glucanases (i.e., it was shielded by other cell wall polymers, presumably pectin).

Compared to previous work done by our group [9,14,33,34], where high dosages of commercial and purified enzymes were added to grape tissues, this work represent a scenario where relatively low levels of enzyme (as low as two gene copies per yeast) were available. This work therefore also confirmed that the CoMPP method is sensitive enough to spot small changes in the cell wall, which were not detectable in the characteristics of the wine produced by these yeasts (unpublished results). Furthermore, previous studies that aimed to demonstrate the effect of the EPG activity produced by PR7 on wine characteristics showed an increase in aroma compounds for *V. vinifera* cv Shiraz wine produced with this yeast, but also a decrease in polyphenol levels (with the exception of quercetin) [26]. The authors speculated that the polyphenols were caught in the thick sediment visible in the PR7 wines. In other wine-making trials with PR7, filter sterilised wine sampled at the end of the fermentation was shown to contain polygalacturonase activity as tested in a plate assay (data not shown), but no significant increase in colour density or total phenolics (compared to wine fermented by the parent strain VIN13) was visible for PR7 (unpublished results from wine making trials at SAGWRI). It is interesting to see that, despite these previous results, we were able to detect the effect of the EPG hydrolysis on the cell walls of grape berries by using CoMPP analysis (Figure 3, Figure 5 and Figure 6). It is likely that the buffered conditions of the fermentations (pH 5.7) conducted in this study (in contrast with wine pH (±3.5)) played a crucial role in producing these results.

The degradation and disappearance of partially methylesterified HG labelling (mAbs LM18 and LM19) in both PR7 samples might be an indication that this yeast strain has pectin methylesterase activity in addition to the EPG activity, although there is no evidence for this in the literature. Furthermore, the study by Eschstruth and Divol [26] did not report an increase in methanol concentration for wine fermented by the PR7 or *S. paradoxus* R088 strains.

With the addition of the *end1* gene, the resulting PR7 END1 strain shows an enhanced effect, suggesting a synergistic action between the two enzymes, thus repeating the effects previously observed [9] between EPG and EG. In the samples fermented with PR7 END1, depectination took place and cell wall polymers were unravelled due to presumed partial degradation. This possibly facilitated better access for the extraction solutions to extract more cell wall polymers, thus rendering an enhanced signal for mAbs such as RG-I, RG-I side chains and cell wall proteins. This mild enzyme hydrolysis was thus only breaking up the cell wall polysaccharides into oligosaccharides, which were not small enough to exit the cell wall environment unless aided by an extraction solvent. Mild enzyme hydrolysis can in fact cause the development of a more porous cell wall and a consequential increase in cell wall surface. These are conditions conducive to the increased binding of proanthocyanidins to the cell wall polymers [35,36]. Thus, when PR7 and PR7 END1 were previously used in wine fermentations, the enzymes that were produced might have done some minor damage to the cell walls of the grape berries. This might have been just enough to form numerous new pores in the cell wall, and therefore enable proanthocyanidins that were released during the crushing and fermentation of the grapes to bind to this newly enlarged surface area [36]. This can lead to a lower proanthocyanidin concentration in the wine and thus fewer stabilising factors for anthocyanins to bind, resulting in wine with lower colour stability and perhaps also a lower colour density. Similar outcomes were reported in studies performed on apple pomace [37], where it was found that proanthocyanidin binding to cell walls during mechanical disruption of fruit led to limited pectin depolymerisation, meaning only low levels of polysaccharide polymers could be extracted from the tissue. The authors speculated that the presence of the proanthocyanidins restricted the access of polysaccharide degrading enzymes to the cell wall polymers. These authors also saw an increase in pectins that were extracted in the NaOH fraction of apple pomace and they associated this with the proanthocyanidins bound to the cell wall polymers [37]. This corresponded with our results, where the composition of the NaOH extracts was dominated by pectin polymers (RG-I and RG-I side chains). PR7 END1 samples showed the largest increase in these polymers and extracted HG (mAb LM19), which can substantiate the pore-forming activity linked to the enzymes produced by this yeast.

## 5. Conclusions

With this study, we evaluated cell wall profiling techniques, in particular, the CoMPP method, for the detection of cell wall changes in grape tissue caused by hydrolytic enzymes expressed by yeast strains. We found that CoMPP is a highly sensitive method, which can detect the effect of relatively low enzyme quantities on the grape pomace cell walls. The results led to insights into the structure of the cell wall of grape pomace and how different enzymes act on their own or in combination to unravel the cell wall polymers. It also provided hints as to the possible interactions between enzymes, cell wall polymers and grape polyphenols when they come into contact after grape berry cell disruption during processing (e.g., wine making). The yeast strains produced in this study expressed the enzymes (both native and recombinant) while using the grape tissue as a growth medium and the study serves as a proof of concept for the implementation of similarly engineered strains in the cost-effective processing of grape pomace.

It would be interesting to determine the extent of the role played by the buffered conditions in the results obtained. It would be possible to adjust the pH of pomace in the pomace valorisation industry, though this would entail additional cost. Alternatively, another endo-glucanase with a suitable optimum pH could be integrated into the yeast genome. Furthermore, it would be interesting to determine if there is a critical level of enzyme hydrolysis that would ensure the release of revenue-generating compounds such as polyphenols.

The cell wall profiling methods (CoMPP and monosaccharide analysis) used in this study enabled us to show how hydrolytic enzymes produced by fermenting yeast strains modify the cell walls of grape berries and will be very valuable tools for the evaluation of the capacity of any microorganism, isolated or engineered, to change plant cell wall composition.

## Figures and Tables

**Figure 1 biology-11-00664-f001:**
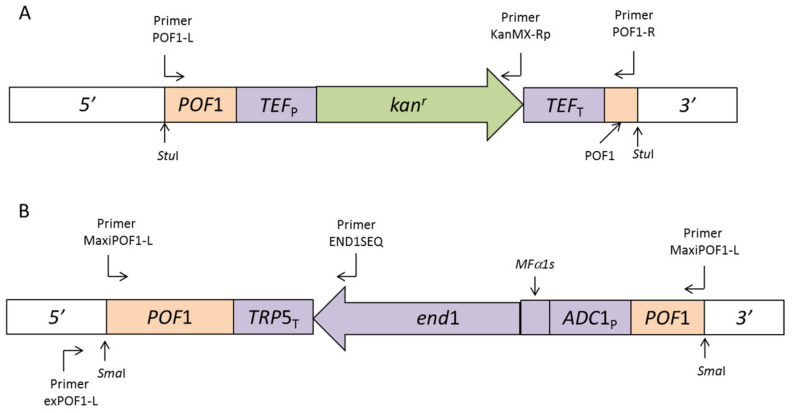
(**A**,**B**) Schematic presentation of the integration cassettes prepared for the disruption of the *POF1* gene, through the exploitation of the homologous recombination system of yeast, on the genomes of VIN13 and PR7. The primer alignment positions of POF1-L and -R as well as MaxiPOF1-L and -R that were used to amplify the *POF1* homologous regions are indicated on the diagram as well as the primers that were used to confirm the integration event at the intended position on the genome (exPOF1-L, MaxiPOF1-L and END1SEQ).

**Figure 2 biology-11-00664-f002:**
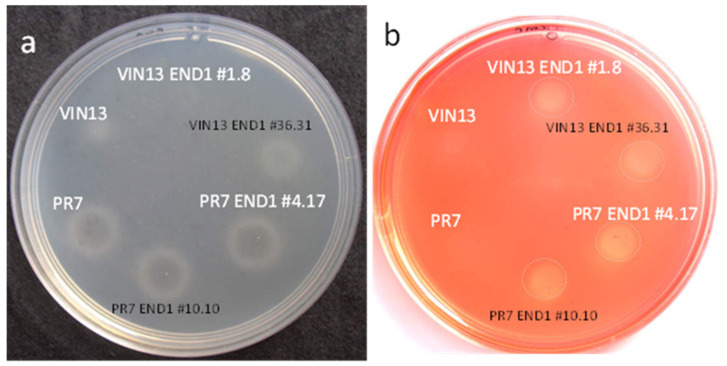
(**a**) Polygalacturonase activity (clear zones inside white rings) on a polygalacturonic acid agarose plate and (**b**) endo-β-1,4-glucanase activity (light yellow zones, circled in dashed lines) on a CMC agar plate with the yeast strains used in this study (VIN13, VIN13 END1 #1.8, PR7, PR7 END1 #4.7). Other strains that were also successfully integrated with the MPOF END1 cassette (VIN13 END1 #36.31 and PR7 END1 #10.10) are also shown here, but they were not used in the pomace fermentations.

**Figure 3 biology-11-00664-f003:**
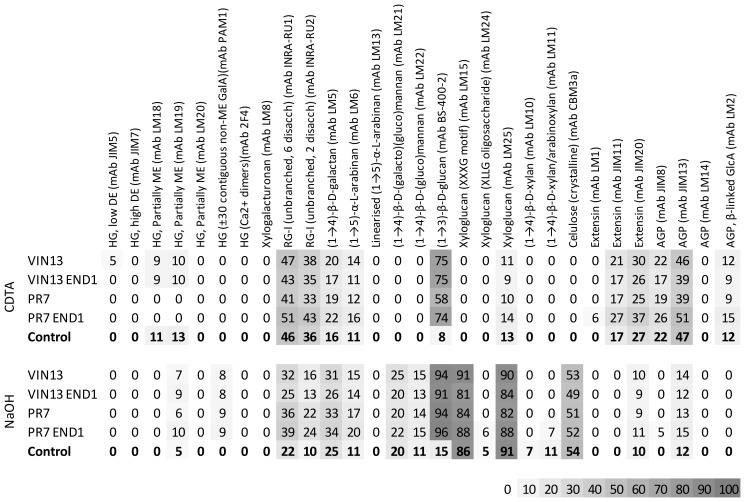
CoMPP data showing the relative abundance of cell wall polymers that were extracted with CDTA and NaOH from AIR. The AIR was isolated from unfermented Chardonnay pomace (Control) and Chardonnay pomace fermented with yeast producing hydrolytic enzymes (V13 END1, PR7 and PR7 END1, as well as VIN13 as a no-enzyme control). The values are the average of four biological samples.

**Figure 4 biology-11-00664-f004:**
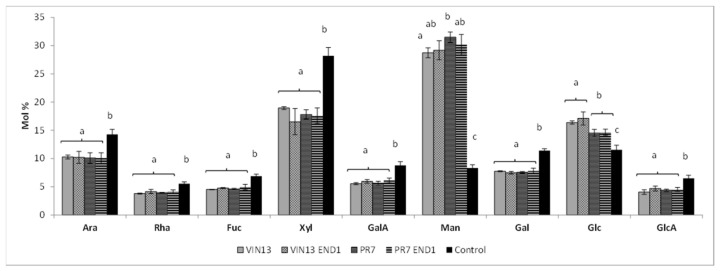
Monosaccharide composition of the cell walls isolated from Chardonnay pomace fermented with different yeast strains. The bars represent the average of four biological and two technical repeats. Different letters indicate a statistically significant difference (95% confidence interval) between the different treatments for a specific monosaccharide. Ara, arabinose; Rha, rhamnose; Fuc, fucose; Xyl, xylose; GalA, galacturonic acid; Man, mannose; Gal, galactose; Glc, glucose; GlcA, glucuronic acid.

**Figure 5 biology-11-00664-f005:**
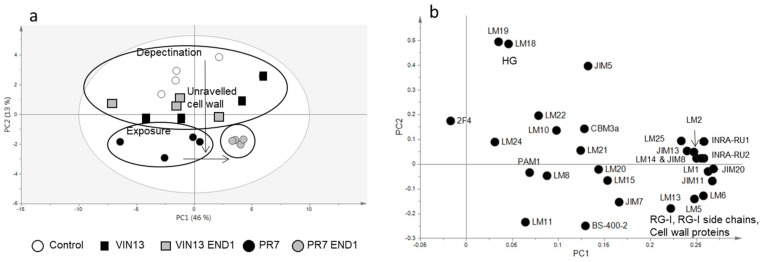
CDTA fraction of the CoMPP data portrayed in a PCA score (**a**) and loading plot (**b**), showing the effect of the yeast fermentations on the pomace cell wall composition. The variable BS-400-2 was excluded from the analysis. The PCA score plot at the top shows the PR7 END1 samples clustering together on the opposite side of the control (unfermented) samples.

**Figure 6 biology-11-00664-f006:**
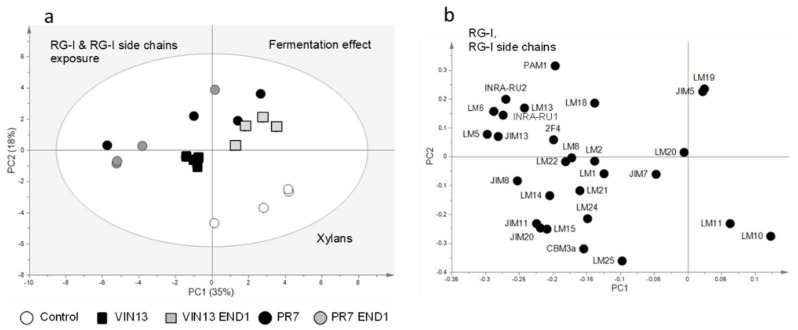
CoMPP data of the NaOH fraction portrayed in a PCA score (**a**) and loading plot (**b**) showing the impact of the hydrolytic enzymes (secreted by the yeast strains during the fermentation) on the pomace cell wall composition. Variable BS-400-2 was excluded from the analysis.

## Data Availability

The data presented in this study are available on request from the corresponding author.

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
