# Peer review of "Commercial Yeast Strains Expressing Polygalacturonase and Glucanase Unravel the Cell Walls of Chardonnay Grape Pomace"

_biology, 2022, doi:10.3390/biology11050664_

Round 1

Reviewer 1 Report

The work is logical, deals with a very current problem of waste - pomace disposal and its further use.

The literature review and methodological part is written in an understandable way, clear and good.

Advanced statistical analysis allows for a good interpretation of the obtained test results

My general remark concerns:

Line 204: The pomace was autoclaved – to  sterilize, destroy the native grape and microbial enzymes.

And pH = 6.

The question is of how much it changes this pomace environment and whether it has an impact on the activity of the tested/evaluated strains. For sure this stage will modify the cell walls of grape berries when compared to unautoclaved material.  Please refer to it at least theoretically.

Sterilization and pH regulation can be difficult in industry conditions, of course possible but expensive.

There is no conclusion part, where general view of the results obtained could be presented and pointed. Such a summary is clearly missing to organize the conclusions.

Reviewer 2 Report

This is a review of the manuscript titled 'Commercial yeast strains expressing a polygalacturonase and glucanase unravel the cell walls of Chardonnay grape pomace'

The manuscript is well written and brings insights on the activities taking place in a valorised material like grape pomace using genetically modified yeast strains. The following issues pointed out below will improve the manuscript.

Line 25: Replace 'microorganisms' with 'yeasts'.

Line 41: Replace 'microorganisms' with 'yeasts'.

Check the rest of the manuscript. the use of a broad term like 'microorganisms' severally when 'yeasts' ought to have been used is supposition and erroneous. Apart from E.coli competent cells used for electroporation, only yeasts were analysed. No bacteria or mould was used for the fermentation of grape pomace in this study.

A brief description of some methods used and more information are required for the supplementary tables. Where modification was carried out, the explicit step introduced or removed and/or what was actually carried out should be stated to allow others to reproduce the work.

Avoid quoting Sambrock only for methods. This should be supported with a brief discussion to enable others to reproduce the work exactly. the information from the manufacturers of kits and reagents may be more beneficial.

What is the reference for Lambda DNA cut with restriction enzymes EcoRI and HindIII?

Line 90: Authors refer to Table A1. There is no appendix. I assume the authors mean Table S1 in the supplementary files.

In table S1, how will reference 15 help other workers to source PR7 plasmid?

Ref 15 was not even written as per the journal's guide (Correct it). More information should be provided on the source of the PR7 plasmid.

In the result and discussion section, it is not clear what polyphenol of commercial interest (if any) was produced. Include specific polyphenols expected in the introduction and discuss whether viable commercial quantities were produced.

Table S2: Were these primers designed by the authors? If yes, the 'this study' column should be added. If not, the reference should be provided. 

 Authors concluded ............ will be a very valuable tool for the evaluation of the capacity of any microorganism, isolated or engineered, to change the plant cell wall composition.......... This is certainly not correct. the capacity for lactobacillus. Pichia and Candida's species have not been tested. Again, replace 'microorganisms' with 'yeasts'. or even Saccharomyces.
